# Associating Functional Neural Connectivity and Specific Aspects of Sensorimotor Control in Chronic Stroke

**DOI:** 10.3390/s23125398

**Published:** 2023-06-07

**Authors:** Adam Baker, Christian Schranz, Na Jin Seo

**Affiliations:** 1Department of Health Sciences and Research, College of Health Professions, Medical University of South Carolina, 77 President St., Charleston, SC 29425, USA; bakerdon@musc.edu (A.B.); schranz@musc.edu (C.S.); 2Division of Occupational Therapy, Department of Rehabilitation Sciences, College of Health Professions, Medical University of South Carolina, 151B Rutledge Ave., Charleston, SC 29425, USA; 3Ralph H. Johnson VA Health Care System, 109 Bee St., Charleston, SC 29425, USA

**Keywords:** stroke, rehabilitation, upper extremity, EEG connectivity, biomechanics, precision medicine

## Abstract

Hand sensorimotor deficits often result from stroke, limiting the ability to perform daily living activities. Sensorimotor deficits are heterogeneous among stroke survivors. Previous work suggests a cause of hand deficits is altered neural connectivity. However, the relationships between neural connectivity and specific aspects of sensorimotor control have seldom been explored. Understanding these relationships is important for developing personalized rehabilitation strategies to improve individual patients’ specific sensorimotor deficits and, thus, rehabilitation outcomes. Here, we investigated the hypothesis that specific aspects of sensorimotor control will be associated with distinct neural connectivity in chronic stroke survivors. Twelve chronic stroke survivors performed a paretic hand grip-and-relax task while EEG was collected. Four aspects of hand sensorimotor grip control were extracted, including reaction time, relaxation time, force magnitude control, and force direction control. EEG source connectivity in the bilateral sensorimotor regions was calculated in α and β frequency bands during grip preparation and execution. Each of the four hand grip measures was significantly associated with a distinct connectivity measure. These results support further investigations into functional neural connectivity signatures that explain various aspects of sensorimotor control, to assist the development of personalized rehabilitation that targets the specific brain networks responsible for the individuals’ distinct sensorimotor deficits.

## 1. Introduction

Stroke is a leading cause of morbidity and a leading cause of long-term disability. In the United States, nearly 800,000 Americans experience a stroke annually [1]. Moreover, on the global scale, there were greater than 100 million strokes in 2019, accounting for the collective loss of approximately 143 million years of full health [2]. Stroke survivors often experience a reduction in sensorimotor function to one side of the body, known as hemiparesis. Hemiparesis commonly presents in an upper extremity [3], restricting the ability of an individual to perform activities of daily living for self-care, vocation, and recreation, thereby reducing quality of life [4,5]. Moreover, many stroke survivors experience hemiparesis for the remainder of their lives even after a full standard course of rehabilitation treatment [6], further emphasizing the need to improve sensorimotor rehabilitation outcomes after a stroke.

Because of the heterogeneity of stroke, precision rehabilitation is a promising avenue to improve sensorimotor recovery outcomes. To achieve precision rehabilitation, a more comprehensive understanding of the neurophysiology underlying impaired upper extremity movement after stroke is required. Thus, investigations into the underlying neurophysiology have taken place as follows. First, it has been shown that the lesion volume alone does not explain the extent of impairment [7,8]. Instead, previous investigations indicate that corticospinal tract integrity is an important predictor of motor recovery after stroke. This conclusion was drawn from studies utilizing Transcranial Magnetic Stimulation (TMS) and diffusion tractography from MRI [8,9,10,11,12]. Using TMS, one is able to determine the presence of a motor evoked potential (MEP) in a given muscle, recorded via electromyography (EMG) [13]. A positive MEP response indicates an intact corticospinal tract, which positively correlates with motor function [14] and is a strong predictor of motor recovery at 3 months post-stroke [13]. On the other hand, investigations using diffusion tractography assess white matter integrity using the diffusivity of water in relation to a myelinated axon [15]. In this way, higher water diffusion along the myelinated axon, often quantified as fractional anisotropy, ref. [15] indicates greater white matter structural integrity, positively correlating with motor function in the chronic phase [16] of stroke recovery and is a predictor of motor recovery at 3 months after stroke [17]. Additionally, it has been shown that neuronal rewiring after stroke, as assessed by both structural and functional neural connectivity, is a prominent contributor to sensorimotor recovery after stroke, beyond lesion location [18,19,20,21,22,23,24]. Taken together, the use of these methodologies provides valuable information to stratify stroke survivors based on individual characteristics [13]. The ability to appropriately classify stroke survivors into subgroups based on these individual characteristics will likely contribute to the development of precision rehabilitation to enhance recovery outcomes post-stroke.

However, past studies have commonly employed clinical motor function assessment scores to quantify impairment. Clinical motor function assessments, such as the Wolf Motor Function Test (WMFT) [25,26] or Fugl–Meyer Assessment of the Upper Extremity are valid and reliable [27,28,29], but are limited in that they do not identify specific contributors to movement deficits, such as slowed muscle contractions [30], the impaired ability to grade force output [31], or impairments in adjusting motor output per afferent inputs [32]. In other words, standardized clinical motor function assessments do not capture the heterogeneity of sensorimotor impairment patterns among stroke survivors. Therefore, the previous literature is limited in that it does not elucidate the functional neural connectivity patterns responsible for specific aspects of impaired sensorimotor control in stroke survivors.

For these reasons, the present study aimed to examine specific aspects of sensorimotor control (e.g., the ability to quickly relax the muscles of the paretic hand or apply an appropriate amount of force with the paretic thumb and index finger) to objectively capture different aspects of sensorimotor impairments and investigate their associations with functional neural connectivity in stroke survivors. Electroencephalography (EEG) was used to examine functional neural connectivity and neural communication [33], as EEG is an effective tool in both neurotypical individuals and stroke survivors to understand the neural connectivity underlying sensorimotor performance [34,35,36,37]. The current study hypothesized that specific aspects of sensorimotor impairment would be associated with distinct functional neural connectivity features.

## 2. Methods

### 2.1. Participants

Twelve chronic stroke survivors with mild-to-moderate upper extremity sensorimotor impairment participated. Chronic stroke survivors were defined as those who had a stroke more than 6 months ago at the time of study enrollment [38]. Mild-to-moderate sensorimotor impairment was defined as scoring 30–60/66 on the Upper Extremity Domain of the Fugl–Meyer Assessment of Motor Recovery after Stroke. The Upper Extremity Domain of the Fugl–Meyer Assessment of Motor Recovery after Stroke evaluates joint reflexes, abnormal synergy patterns, functional mobilities, and coordination for a composite score of upper extremity impairment level [39]. Paretic upper extremity function was assessed using the WMFT and the Box and Block Test (BBT). The WMFT quantifies the time required for participants to perform 15 functional tasks with their paretic upper extremity, such as lifting a pencil from a table, lifting a soda can near the face to simulate drinking, and stacking three checkers on top of one another [25,26,27]. The BBT measures the number of blocks that a participant is able to move from one side to the other side in 1 min [40]. Exclusion criteria included (1) the inability to follow instructions, and (2) a botulinum toxin injection within 3 months before enrollment. The study was approved by the local Institutional Review Board at the Medical University of South Carolina. Written informed consent was obtained from all the participants prior to their participation in the study protocol.

### 2.2. Procedure

#### 2.2.1. Grip Control Data Acquisition

Hand sensorimotor grip control and EEG functional neural connectivity were assessed simultaneously for each participant. Participants performed a grip-and-relax task using the thumb and index finger of the paretic hand against two 6-axis load cells (Mini40, ATI Industrial Automation Inc., Apex, NC, USA) while sitting in front of a computer screen with the arms rested on armrests. For cues, the words “grip” and “rest” were visually presented via computer screen through a custom LabVIEW program (National Instruments Corp., Austin, TX, USA). The grip cues lasted 2 s, followed by 5–6 s rest cues, and this cycle was repeated 100 times. Immediately preceding the data collection, participants practiced gripping at the prescribed force level of 4 N with visual feedback provided by a force meter on the computer screen. This visual feedback was subsequently removed during the data collection to measure the participants’ ability to recreate the target force level using proprioception [31]. The force level of 4 N was decided upon because 4 N is small enough for stroke survivors with moderate impairments to achieve without excessive fatigue, while being distinctive from rest [36].

#### 2.2.2. EEG Acquisition

During the grip-and-relax task, EEG was also recorded in the BrainVision Recorder software using a 96-channel actiCAP and BrainAmp MR plus amplifier (BrainVision LLC, Morrisville, NC, USA). The 10–20 international system was used to position electrodes, with Cz at the apex of the head, the ground electrode at AFz, and the reference electrode at FCz. The EEG data were recorded at 1 kHz after the application of a 0.1–200 Hz bandpass filter and a 60 Hz notch filter. The timing of the grip and rest cues from the LabVIEW program were also recorded with the EEG data to enable data synchronization and event-related data analysis.

#### 2.2.3. MRI Acquisition

To enable EEG source localization, a structural T1-weighted brain MRI was obtained in isometric 1 mm^3^ voxel sizes through the MPRAGE sequence [41] using a Siemens Prisma 3T TIM Trio MRI scanner (Siemens AG, Munich, Germany). In total, 2 of the 12 participants had contraindications to MRI, therefore their MRI could not be obtained. To characterize lesion locations among participants, lesions were manually drawn using MRIcron [42], verified by a stroke neurologist, and normalized to the Montreal Neurological Institute [43] space. The lesion locations for the participants who underwent an MRI are summarized in Figure 1.

### 2.3. Analysis

#### 2.3.1. Sensorimotor Grip Control

From the paretic hand grip-and-relax task, the following sensorimotor control measures were obtained, (1) reaction time, (2) relaxation time, (3) force magnitude control, and (4) force direction control (Figure 2). Reaction time is the elapsed time between grip cue onset and grip commencement [44]. Grip commencement was defined as when the force level exceeded the mean plus 3 times the SD of the force level during the rest period [44]. Relaxation time is defined as the duration between rest cue onset and grip termination [30]. Grip termination was defined as when the force level became below the mean plus 3 times the SD of the force level during the rest period [30]. Force magnitude control was calculated as the difference between the average grip force and the prescribed force (Figure 2A) [31]. Force direction control was quantified as the mean angular deviation of digit force from the direction normal to the grip surface during grip (Figure 2B) [45,46]. Each of these four grip measures were averaged over the repetitions for each participant.

#### 2.3.2. Brain Connectivity

For brain network connectivity, the EEG data were pre-processed as described in previous studies [24,36]. In short, the EEGLAB [47] toolbox within MATLAB (The MathWorks, Inc., Natick, MA, USA) was used. EEG data were band-pass filtered using a 0.5–50 Hz Butterworth filter. Independent component analysis was implemented for the removal of artifact sources, such as eye blinks, via the ADJUST algorithm [48]. Trials were identified and rejected from further analysis if (1) a grip was missed, as measured by the force readings of sensors during the grip phase, or (2) if EEG data values were lesser than −450 µV or greater than 450 µV peak-to-peak amplitude. On average, 87 of the 100 attempted trials (SD = 18 trials) for each participant remained for further analysis.

Lesion-specific source modelling was performed using the patient’s T1-weighted MRI structural brain scan. Cortical surfaces were reconstructed, then segmented in FreeSurfer [49] and subsequently imported into Brainstorm [50]. Cortical surfaces from FreeSurfer were modeled within Brainstorm using 15,000 vertices. For the two participants with MRI contraindications, the Montreal Neurological Institute average brain [43] was used in place of the participant’s MRI. The pre-processed EEG data were imported and co-registered, and a boundary element head model using the OpenMEEG method [51] was then created within Brainstorm. EEG sources were computed using the minimum norm estimate [52]. Functional neural connectivity was assessed as imaginary coherence to control for volume conduction and field spread artifacts [53].

The regions of interest included the bilateral sensorimotor cortices, namely the primary motor cortex, primary somatosensory cortex, and premotor cortex. The Desikan–Killiany atlas [54] was used for analysis. However, for five of the participants, the Desikan–Killiany atlas could not be used because of poor segmentation results. Therefore, the sensorimotor regions of interest were manually drawn for these five participants. These sensorimotor brain regions were chosen as regions of interest because they have been shown to be involved in motor task planning, execution, and overall motor function [55,56,57]. Functional neural connectivity within the alpha (α, 8–12 Hz) and beta (β, 13–30 Hz) frequency bands was examined, as they have been demonstrated as prominent during movement planning [58] and movement execution [59], respectively. Lastly, to capture the temporal dynamics of EEG relevant to paretic hand grip performance, functional neural connectivity was examined for both the preparation and execution phases of grip, as defined by the 1 s period before and after the grip cue, respectively.

#### 2.3.3. Statistical Analysis

The connectivity measure that significantly explained each grip measure was identified using regression. Connectivity measures for each frequency band and each grip phase averaged within the lesioned hemisphere, within the non-lesioned hemisphere, and between hemispheres were initially considered, and individual region pairs were further considered as necessary. As an additional analysis, lesion volume was computed as percent volume in the normalized brain and considered as a covariate in the regression model. A significance level of 0.05 was used. Multiple comparisons were not adjusted for because this is a pilot study to generate new hypotheses for future studies with appropriate sample sizes. All statistical analyses were performed using SPSS 27 (IBM, SPSS Inc., Chicago, IL, USA). Lastly, brain connectivity measures that were found significantly associated with each grip measure were visualized using BrainNet Viewer [60].

## 3. Results

### 3.1. Participant Characteristics

The participants averaged a score of 47.7 (standard deviation [SD] = 8.0) on the Upper Extremity domain of the Fugl–Meyer Assessment of Motor Recovery after Stroke [39]. The mean time since stroke of the 12 participants was 61.5 months (SD = 58.2 months), while the average age was 62.3 years (SD = 8.4 years). The demographic information of each participant, as well as their paretic upper extremity functional status assessed using the WMFT [25,26] and BBT score [40] are presented in Table 1.

### 3.2. Relationship between Grip Measures and Functional Neural Connectivity

The main finding of this study is that each grip measure was found to be associated with a different functional neural connectivity measure. Reaction time was found to be strongly associated with α band connectivity within the non-lesioned hemisphere during grip preparation (r = −0.61 and *p* = 0.035 for averaged connectivity within the non-lesioned hemisphere). The respective individual region pair correlation coefficients for each measure are depicted in Figure 3. The correlation coefficients for the individual region pairs were r = −0.59 between the premotor and primary somatosensory cortices, r = −0.58 between the premotor and primary motor cortices, and r = −0.61 between the primary somatosensory and primary motor cortices. Relaxation time was strongly associated with β band connectivity within the lesioned hemisphere during the grip execution phase (r = 0.75; *p* = 0.007; Figure 3B). The correlation coefficients for the individual region pairs were r = 0.71 between the premotor and primary somatosensory cortices, r = 0.83 between the premotor and primary motor cortices, and r = 0.59 between the primary somatosensory and primary motor cortices.

Force magnitude control was found to be strongly associated with α band connectivity within the lesioned hemisphere during the grip preparation (r = 0.66; *p* = 0.021; Figure 3C). The correlation coefficients for the individual region pairs were r = 0.47 between the premotor and primary somatosensory cortices, r = 0.63 between the premotor and primary motor cortices, and r = 0.81 between the primary somatosensory and primary motor cortices. Force direction control was found to associate with α band connectivity between the non-lesioned premotor and primary somatosensory cortices during grip execution, though this did not reach statistical significance (r = −0.57; *p* = 0.051; Figure 3D).

When lesion volume was considered as a covariate, the connectivity that was most strongly associated with each grip measure did not change. The lesion volume was not found to be significantly associated with the first three grip measures. However, for force direction control, the addition of lesion volume led to the finding that force direction control was significantly explained by both the α band connectivity between the non-lesioned premotor and primary somatosensory cortices (*p* = 0.003) and lesion volume (*p* = 0.008). This interplay is depicted in Figure 4. The force direction control was worse (higher) when the α band connectivity was low and the lesion volume was large, but not when either the α band connectivity was relatively high or the lesion volume was relatively small.

As a negative control, regression coefficients between connectivity measures and clinical motor scores were also examined. While connectivity was significantly associated with the four grip measures, connectivity was not significantly associated with clinical motor scores of BBT and WMFT (*p* > 0.065). This finding indicates that functional neural connectivity measures are associated with the detailed sensorimotor grip control measures better than the gross clinical motor scores.

## 4. Discussion

### 4.1. Overall Findings

Here, it was found that specific aspects of sensorimotor control were associated with distinct functional neural connectivity in chronic stroke survivors. The present results contribute to the growing number of studies utilizing biomechanical data and functional neural connectivity to investigate motor recovery after stroke [61,62,63]. Importantly, the current study is the first, to the authors’ knowledge, to directly compare the relationships between functional neural connectivity and specific aspects of sensorimotor control in chronic stroke survivors. Significant associations were observed between functional neural connectivity and specific aspects of sensorimotor control, but not between functional neural connectivity and standardized clinical motor function tests (WMFT and BBT). These current data support the notion that a more comprehensive understanding of the neurophysiological correlates underlying sensorimotor impairments after stroke may be obtained when specific measures of sensorimotor control are studied in relation to functional neural connectivity. Thus, the current study highlights the role of specific sensorimotor control measures, in addition to clinical motor function scores, to elucidate impairment mechanisms after stroke.

### 4.2. Connectivity and Sensorimotor Function

Reaction time was found to hold an inverse correlation with α band connectivity in the non-lesioned hemisphere, especially between the primary motor cortex and primary somatosensory cortex, during grip preparation (Figure 3A). The inverse correlation suggests a higher connectivity value during the movement preparation phase within the α band is associated with faster reaction time performance. These current data are in accordance with findings from previous studies demonstrating α frequency band activity immediately preceding movement in the sensorimotor cortex contralateral to the moving hand [64] and the lessened extent of α frequency band activity change after stroke [65]. Moreover, these data corroborate the results of previous studies in which the α band is known for its role during cognitive tasks and attention [58], and reaction time may reflect the attentiveness for grip [66].

Relaxation time was found to hold a positive association with β band connectivity within the lesioned hemisphere during grip execution (Figure 3B), supporting previous findings suggesting the high prominence of β band rhythms during sensorimotor activity [67] and motor learning [24,68]. Relaxation time may be considered a measure of feedback control [69], reflecting the ability for one to rapidly cease muscular contractions of the intrinsic and extrinsic muscles of the paretic hand and forearm, respectively, even in the presence of excessive muscular tone after stroke [30,70,71]. The current results indicate that higher connectivity is associated with longer relaxation time durations with an impaired ability to quickly terminate a muscle contraction (i.e., worse performance). This negative effect of higher functional neural connectivity in the lesioned hemisphere may represent maladaptation possibly associated with impaired inhibitory control following stroke [72,73].

Force magnitude control in the paretic hand was found to associate with the α frequency band within the lesioned hemisphere prior to grip (Figure 3C). It has been established that stroke survivors have deficits in their ability to grade force output during an object manipulation task in the paretic hand [74,75,76]. The present study suggests the involvement of the lesioned hemisphere’s attention-related functional network for modulating force magnitude.

Force direction control was significantly explained by both α connectivity between the non-lesioned premotor cortex, the primary somatosensory cortex, and lesion volume (Figure 4). Significance was obtained only upon the addition of lesion volume as a covariate. These results suggest that increased connectivity or residual brain resources could compensate for each other to better direct force application (i.e., a smaller angle between the direction of force application and normal direction).

Interestingly, two of the four grip measures were significantly associated with brain connectivity within the lesioned hemisphere, while the other two grip measures were significantly associated with brain connectivity within the non-lesioned hemisphere. The current results did not observe interhemispheric connectivity to be significantly associated with any of the grip measures. These findings may indicate how lesion location leads to functional deficits to somatosensation [77,78], motor performance [79,80], and subsequent recovery outcomes [81]. Heightened involvement or activity of the non-lesioned hemisphere relative to the lesioned hemisphere may indicate either the presence of compensatory mechanisms or bilateral drive needed to control paretic arm movement after stroke [82,83,84,85].

### 4.3. Implications

In summary, the current investigation is the first to the authors’ knowledge aimed to understand the neural underpinnings of force output control in the paretic hand in chronic stroke patients. Regarding methodologies, the use of specific biomechanical measurements of hand grip performance appeared to result in clearer associations with functional neural connectivity, compared to use of conventional clinical assessment scores. In addition, while EEG has been shown to be a potentially useful biomarker for recovery after stroke [86,87], the present study adds that functional neural connectivity may also elucidate sources of different aspects of motor impairment. Functional neural connectivity assessed by EEG may depict functionally relevant changes in neural states and may complement knowledge obtained from structural connectivity [62,88]. Taken together, developing an enhanced understanding of sensorimotor impairments after stroke in this way will build a stronger scientific premise to underlie the development and implementation of precision rehabilitation for sensorimotor recovery after stroke. For instance, personalized treatments using neuromodulation may be strategically implemented to alter the activity of the functional neural network or brain region deemed responsible for an impaired aspect of sensorimotor control, thereby facilitating precision rehabilitation for improved rehabilitation outcomes.

### 4.4. Limitations

While specific functional neural connectivity was suggested for each grip behavior, due to the small sample size in the current cohort, these data should be taken with caution and used only as a starting point for further investigations. Ongoing studies seek to provide stronger, more robust evidence to elucidate the relationships between specific measures of sensorimotor performance and functional neural connectivity [89]. Second, though the majority of stroke occurrences are within older populations, there is a growing number of younger individuals experiencing a stroke [1]. The current cohort of stroke survivors did not include younger stroke survivors, as the youngest participant was 46 years of age. Thus, variables such as natural aging may be present in the current dataset that would not otherwise be present in a younger cohort. Lastly, absent in these data is the presence of 3-dimensional coordinate values for each of the 96 EEG electrodes. While the researchers followed the best practice for placing the EEG cap, slight variations in electrode locations may have been present, which went unaccounted for.

## 5. Conclusions

In the present study, specific measures of sensorimotor control of the paretic hand were found to associate with functional neural connectivity to a greater degree than standardized motor function tests in chronic stroke survivors. While these findings should be taken with caution due to the small sample size, this work provides a proof-of-concept demonstration supporting the notion that prevalent standardized clinical motor function tests are not comprehensive and that biomechanical analyses can provide a more complete picture of motor deficits with specific neural correlates after stroke. Lastly, these data support further investigations to understand the neurobiology of upper extremity sensorimotor control after stroke. Understanding the relationships between functional neural connectivity and upper extremity sensorimotor control will assist in the development of the scientific premise to underlie precision sensorimotor rehabilitation. These personalized paradigms, such as neuromodulation, may be able to target specific functional networks responsible for a given aspect of sensorimotor control.

## Figures and Tables

**Figure 1 sensors-23-05398-f001:**
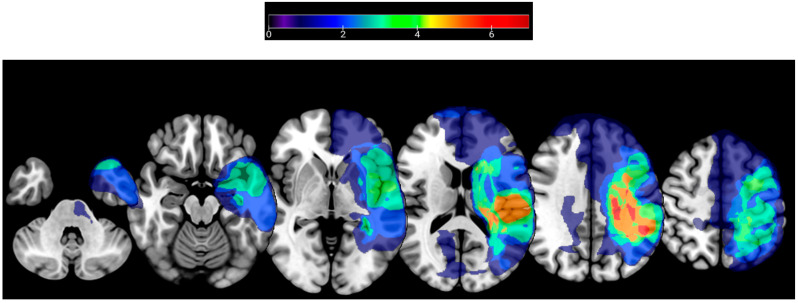
Lesion locations for participants that underwent structural T1-weighted MRI. The color bar indicates the number of participants with lesions at a given location.

**Figure 2 sensors-23-05398-f002:**
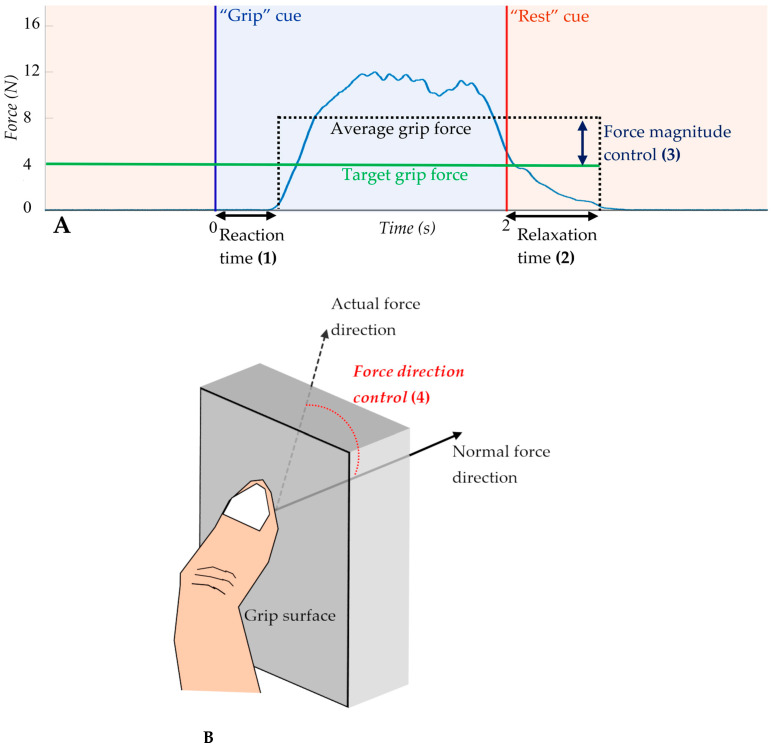
Graphical representation of (1) reaction time (2) relaxation time, and (3) force magnitude control extracted from the hand grip-and-relax task (**A**). Schematic representation of force direction control (4), calculated as the average angle of force application during grip (**B**).

**Figure 3 sensors-23-05398-f003:**
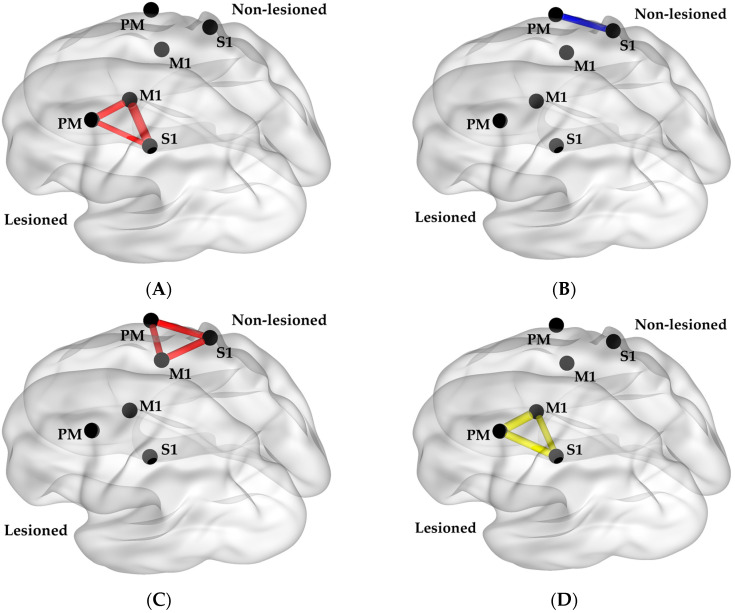
Brain connectivity measures that were found to have significant correlations with (**A**) reaction time, (**B**) relaxation time, (**C**) force magnitude control, and (**D**) force direction control. A thicker line indicates a stronger absolute value of the correlation coefficient (r). Red indicates the alpha band during the preparation phase, blue indicates alpha band during the execution phase, and yellow indicates beta band during execution phase. M1 = Primary Motor Cortex; PM = Premotor Cortex; S1 = Primary Somatosensory Cortex.

**Figure 4 sensors-23-05398-f004:**
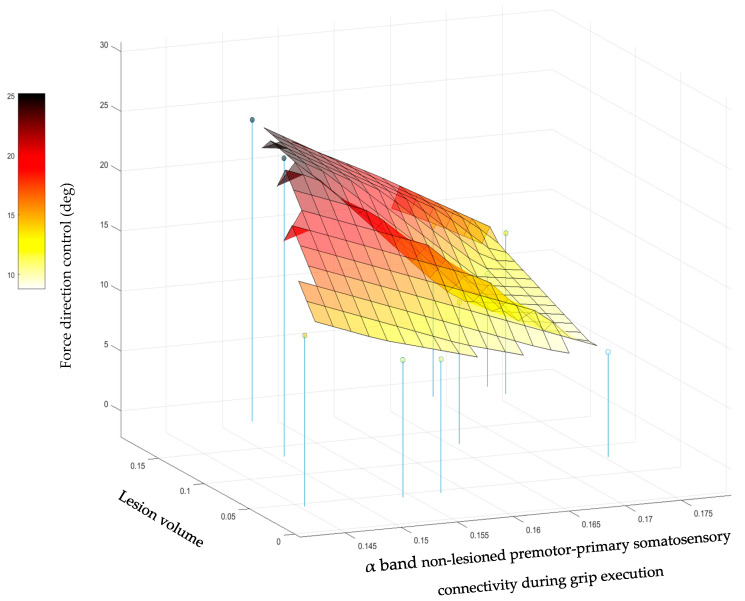
The relationship describing the way force direction control was associated with the lesion volume (in percent) and connectivity between the non-lesioned premotor cortex and primary somatosensory cortex in the alpha frequency band during grip execution.

**Table 1 sensors-23-05398-t001:** Demographic data of participants.

Study ID	Age	Sex	Dominant Hand	Affected Upper Extremity	Time Since Stroke (Months)	Fugl-Meyer Upper Extremity Score (/66)	Wolf Motor Function Test Time (Seconds)	Box and Block Test Score
1	58	F	R	R	75	57	3.5	46
2	61	M	R	R	23	41	29.8	12
3	63	F	R	R	65	58	5.0	31
4	46	F	R	R	17	35	33.2	12
5	64	M	R	R	39	59	2.4	50
6	59	F	R	R	27	51	9.7	18
7	58	M	R	R	34	43	14.9	17
8	68	M	R	L	193	53	4.5	35
9	73	F	R	L	15	40	4.6	34
10	66	M	R	R	47	43	6.3	32
11	53	M	R	L	25	39	16.0	24
12	79	M	R	R	178	53	6.6	34

## Data Availability

Data involved in this research may be made available upon reasonable request to the corresponding author.

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
