# Peer review of "Associating Functional Neural Connectivity and Specific Aspects of Sensorimotor Control in Chronic Stroke"

_sensors, 2023, doi:10.3390/s23125398_

Round 1

Reviewer 1 Report

Baker et al., sought to understand the neural underpinnings of impaired grip force control in a group of 12 individuals with chronic stroke. They evaluated connectivity in the alpha and beta bands bilaterally, and found that timing parameters (reaction and relaxation times) were strongly related to alpha and beta-band connectivity variously in the contralesional and ipsilesional hemispheres. The authors conclude that this work demonstrates that potential held by connectivity-based analyses for identifying e.g., potential rehabilitation targets.

Overall, I found this paper well-written and straightforward. I have a few comments/questions for clarification, enumerated below:

1. Were all participants right-handed? It might be good to mention this, and add a simple handedness column to Table 1.

2. Please provide a rationale for not adjusting for multiple comparisons.

3. In-text at lines 260-271, it is mentioned that the correlation coefficients for individual region pairs are shown in Fig 3, however, Fig 3 has no annotations of correlation values. Perhaps either labelling the figure, or adding a small additional table, would be useful to provide more information. Also, presumably, this is referring only to the strongest/most significant correlations as mentioned in-text, e.g., on line 263.

Author Response

Please see attached word document.

Reviewer 2 Report

Interesting and well written study regarding the association between the Functional Neural Connectivity and Specific Aspects of Sensorimotor Control in Chronic Stroke. 

The study is well conceived, methodology is clear and results are well presented. 

Some minor aspects to adjust prior to publication:

Discussion is critical and balanced. As a minor suggestion I would like you to add some lines regarding the importance of the extension of the ischemic area on function outcome and regarding Sensorimotor Control in Chronic Stroke (e.g. doi: 10.1136/jnis-2022-019557).

Conclusions could be better focused on the main findings of the works. 

Author Response

Please see the attached word document.

Reviewer 3 Report

The study was to examine specific aspects of sensorimotor control to objectively capture different aspects of sensorimotor impairments and investigate their associations with functional neural connectivity in stroke survivors. They used EEG to examine functional neural connectivity and neural communication. Twelve chronic stroke survivors with mild-to-modarate UE sensorimotor impariment participanted. Reaction time was found to be strongly associated with a-band connectivity within the non-lesioned hemisphere during group preparation. Force magnitude control was found to be associated with a-band connectivity within the lesioned hemisphere during the grip preparation.  

The study is expected to be a good data for evaluating and training the upper extremity function of stroke patients. I'm grateful for the opportunity to review such a good study. 

Author Response

Please see the attached word document. 

Reviewer 4 Report

This study addresses an important functional association in chronic stroke. A very well written and well presented study. The introduction provides relevant context and background information, highlighting a current gap in evidence and justifying the need for this study, and ongoing work, well. The methods are clear, although it would be worth consideration to move the demographic data to the results section. The results are clearly presented and the addition of images (in both results and methods) service as an important tool for understanding the value of this study. The discussion is high level and provides real world evidence around why theses associations are important and how this information could be utilized to inform rehabilitation practice or patient care overall. You have identified the limitations of this study, although I would add that the sample did not include any one of a younger age group (under 35yo). Although the majority of strokes occur over the age of 65yo, there is a growing proportion of young stroke survivors. Some very minor suggestions below. 

Line 33 - 800,00 Americans experience a stroke, is this per year? 

Line 411 - Remove "here" from the start of your sentence. 

Author Response

Please see the attached word document. 
